# Targeted Removal of Galloylated Flavanols to Adjust Wine Astringency by Using Molecular Imprinting Technology

**DOI:** 10.3390/foods12183331

**Published:** 2023-09-05

**Authors:** Guorong Du, Xiaoyu Wang, Qinghao Zhao

**Affiliations:** 1School of Biological and Environmental Engineering, Xi’an University, Xi’an 710065, China; 93683330@xawl.edu.cn; 2College of Food Engineering and Nutritional Science, Shaanxi Normal University, 620 West Chang’an Avenue, Xi’an 710119, China; 3Engineering Research Center for High-Valued Utilization of Fruit Resources in Western China, Ministry of Education, Xi’an 710119, China

**Keywords:** astringency, model tannin solution, gallotannins, molecular imprinting, electronic tongue

## Abstract

Excessive galloylated flavanols not only cause instability in the wine but also lead to unbalanced astringency. Although clarification agents are always used to precipitate unstable tannins in wine, the non-specific adsorption of tannins results in the failure to precisely regulate the tannin composition of the wine. In this work, molecularly imprinted polymers (MIPs) with template molecules of galloylated flavanols were designed to specifically adsorb gallotannins to reduce wine astringency. The results showed that the “pores” on the surface of the MIPs are the structural basis for the specific adsorption of the target substances, and the adsorption process is a chemically driven single-molecule layer adsorption. Moreover, in the mono/oligomeric gallotannin-rich model solution, the adsorption of gallotannins by I-MIPs prepared as single template molecules reached 71.0%, and the adsorption capacity of MIPs for monomeric gallotannins was about 6.0 times higher than polymeric gallotannins. Given the lack of technology for the targeted adsorption of tannins from wine, this work explored the targeted modulation of wine astringency by using molecular imprinting techniques.

## 1. Introduction

Astringency is described as a pucker, rough, or drying mouth-feel, and the richness of astringency gives red wine a unique style [1,2]. However, in practice, excessive astringency can cause disgust in consumers, and it is desired that some mild astringency sensation remains. The addition of exogenous tannin can correct weak astringency, but the way to correct excessive astringency seems to be through tannin oxidation, aging, and clarifier adsorption [3]. These methods inevitably have a significant impact on the color of the wine whilst reducing its astringency. It is still a problem to specifically reduce the tannins in a short time without affecting other qualities of the wine.

In recent years, many studies have focused on gallotannins, which were believed to have a rougher sensation with increased coarseness, drying, and chalkiness [4]. Schoebel et al. [5] suggested that gallotannins enhance the electrical signals of the trigeminal nerve, leading to a stronger astringency than other flavanols. Epigallocatechin gallate (EGCG) and Epicatechin-3-O-gallate (ECG) are the most common monomer gallotannins in wine, and both of them can format the strong intermolecular network between salivary and proteins [6].

The reduction of tannin astringency includes the use of clarifying agents, the addition of polysaccharides, alkali treatment, gene editing, etc. [7,8,9,10]. These methods all modify the tannin composition to varying degrees, thus greatly changing the taste of the wine. Polysaccharides and adsorbents are commonly used to reduce astringency. However, the problem caused by the use of adsorbents and polysaccharides is that all the fining includes the non-specific adsorption of polyphenols, inevitably affecting other sensory qualities of the wine and even reducing wine quality [11,12,13].

Molecular imprinting polymers (MIPs) provide a new idea to develop a new molecular-targeting material to reduce polyphenols, which contribute to bad taste [14,15,16]. MIPs have attracted great attention since they exhibit the excellent ability of specific recognition and binding toward the target molecules [17,18,19]. The composition of tannins in wine is complex, and the structure and content of tannins vary between varieties, but both monomers and oligomers are present. Gallic acylated tannins, such as (−)-Epicatechin Gallate (ECG), can be involved in the formation of polymeric tannins, and it is a question whether the specific adsorption of acylated tannins by MIPs can also cause a reduction in polymeric gallotannins.

In this work, we prepared MIPs with single and composite templates of monomeric gallotannins. Not only was the adsorption of MIPs on gallotannins investigated, but four groups of tannin solutions with different characteristics were also reconstructed to study the specific adsorption effect of MIPs. The effect of MIPs on the astringency intensity of the model solution was also investigated using fluorescence and the electronic tongue and sensory evaluation. In addition, the effect of MIPs on the color quality of the wine was assessed. The goals of this study were to explore the feasibility of using MIPs to specifically modulate wine astringency and to provide pathways of tannin-targeted regulation for the improvement of wine astringency.

## 2. Materials and Methods

### 2.1. Chemicals

Flavanol monomers (≥95%) including (+)-catechin (C, CID: 9064), (−)-epicatechin (EC, CID: 72276), (−)-epigallocatechin (EGC, CID: 72277), (−)-epicatechin-3-O-gallate (ECG, CID: 107905), and (−)-epigallocatechin-3-O-gallate (EGCG, CID: 65064) were purchased from Sigma-Aldrich (Shanghai, China). Fe_3_O_4_, Span80 (CID: 131736575), Chitosan (CTS, CID: 71853), Acetonitrile (ACE, CID: 6342), ethyleneglycoldimethacrylate (EGDMA, CID: 521006), and azobisisobutyronitrile (AIBN, CID: 6547) were purchased from Haite biotechnology Ltd. (Xi‘an China).

### 2.2. Preparation of Molecularly Imprinted Polymers Material

All three MIPs in this study were prepared using the following methods, as detailed elsewhere [20]. The general procedure of MIPs was divided into two steps, as showed in Appendix A:

(a) Preparation of MICMS: 1.0 g of Fe_3_O_4_ was dispersed into 40 mL of 20% chitosan solution, and then 8 mL of span-80 and 160 mL of liquid paraffin were added and stirred at 600 rpm for 30 min. Slowly, 6 mL of 25% glutaraldehyde solution was added dropwise and quickly stirred (800 rpm) for 2.5 h. Ethanol/acetone/ethanol (1:1:1, *v*/*v*/*v*) was used to wash the production three times, and MICMS was obtained after vacuum drying at 45 °C.

(b) I-MIPs: Function monomers (0.4 mmol MAA) were mixed in a glass vial with porogen (10.0 mL ACE) and the template flavanol monomer (0.1 mmol EGCG). Then, AIBN (15.0 mg), EGDMA (4 mmol), and 200 mg of MICMS were added. The solution was purged with nitrogen for 15 min and heated in a water bath at 60 °C for 24 h by a reflux device. Imprinted polymers were extracted in a Soxhlet apparatus using MeOH/Acetic acid (9:1, *v*/*v*) as a solvent until no more template molecules were detected in the MIPs. Finally, the polymer was rinsed three times with edible alcohol and dried at 45 °C overnight.

II-MIPs: The preparation process of II-MIPs was carried out similarly, except that the template molecule was replaced with 0.1 mmol EGCG: ECG (1:1, mol/mol).

N-MIPs: Non-imprinted polymers were prepared without the appearance of the template [21].

### 2.3. Fourier Transform Infrared Spectroscopy (FTIR)

Fourier transform infrared spectroscopy was used to determine whether the surface of MIPs was successfully modified with target functional groups. FTIR refers to the method described by Zhao et al. [22]. N-MIPs, I-MIPs, II-MIPs, and CTS were mixed with KBr at a ratio of 1:100, respectively, were fully ground with agate, and then were pressed into tablets. The MIPs were subject to Fourier transform infrared spectroscopy (FTIR) analysis in the range of 400 to 4000 cm^−1^ using an Infrared spectrometer Tensor27(Bruker, Billerica, MA, USA).

### 2.4. Environmental Scanning Electron Microscope Analysis

Environmental scanning electron microscopy (ESEM) was used to observe the structure of cavities on the surface of MIPs [23]. The samples of the scanning electron microscope (N-MIPs, I-MIPs, II-MIPs, and CTS) were sprayed with gold in a vacuum environment, and the spraying time was 100 s. The morphology of MIPs was analyzed by environmental scanning electron microscopy (SEM) Quanta 200 (FEI, Hillsboro, OR, USA).

### 2.5. Adsorption Capacity

#### 2.5.1. Adsorption Equilibrium Time and Kinetics Study

Adsorption properties refer to the previous research method [24]. Samples of 30.0 mg of MIPs (N-MIPs, I-MIPs, II-MIPs) were weighed and mixed with 10.0 mL of EGCG (300 mg/L) solution separately and then shaken at 30 °C. EGCG was measured according to Cheng et al. [25]. Reverse-phase HPLC (RP-HPLC) was performed using an Agilent 1260 HPLC system. Sampling was performed at 0 min, 30 min, 60 min, 90 min, 120 min, 150 min, 180 min, 240 min, 300 min, and 360 min, respectively.Q = ((C_0_ − C) × V)/M(1)

The adsorption quantity (Q) was calculated through Equation (1), where C_0_ and C are the initial and the equilibrium concentration of EGCG, V stands for the volume of the solution, and M refers to the mass of the polymer powder.

The quasi-first kinetic model (Equation (2)) and quasi-secondary kinetic model (Equation (3)) were used to determine whether the adsorption of the MIPs on the target molecules was physisorption or chemisorption:(2)$$\mathrm{Quasi}\text{-}\mathrm{first}\;\mathrm{kinetic}\;\mathrm{model}:\;In\left(q_{e}-q_{t}\right)=Inq_{e}-k_{1}t$$
(3)$$\mathrm{Quasi}\text{-}\mathrm{secondary}\;\mathrm{kinetic}\;\mathrm{model}:\;\frac{t}{q_{t}}=\frac{t}{q_{e}}+\frac{1}{k_{2}{qe}^{2}}$$
where *q_e_* and *q_t_* denote the amount of EGCG adsorbed by MIPs at equilibrium and at time *t*, respectively; *k*_1_ denotes the quasi-primary kinetic adsorption rate constant; *k*_2_ denotes the quasi-secondary kinetic adsorption rate constant.

#### 2.5.2. Isothermal Adsorption and Adsorption Model Analysis

A volume of 10 mL of EGCG solution (50 mg/L, 100 mg/L, 150 mg/L, 200 mg/L, 250 mg/L, 300 mg/L,350 mg/L, 400 mg/L,450 mg/L, 500 mg/L) was mixed with 30 mg of MIPs at 30 °C. After four hours, the MIPs were separated in the presence of a magnetic field, and the supernatant was taken for detection. The Langmuir isothermal adsorption model (Equation (4)) and the Freundlich isothermal adsorption model (Equation (5)) were used to explore whether this MIP material was consistent with monolayer or multilayer adsorption. The model fitting equation was based on the method of Umpleby et al. [26], and the corresponding equation is shown as follows:(4)$$\mathrm{Langmuir}\;\mathrm{isothermal}:\;\frac{1}{Q_{e}}=\frac{1}{Q_{m}K_{d}}\times \frac{1}{C_{e}}+\frac{1}{Q_{m}}$$
(5)$$\mathrm{Freundlich}\;\mathrm{isothermal}:\;Qe=K_{f}{Ce}^{1/n}$$
where *C_e_* is the equilibrium concentration of EGCG in the aqueous solution (mg/L), *K_d_* is the Langmuir adsorption constant (L/mg), *Q_m_* is the maximum adsorption capacity of the adsorbent (mg/g), and *Q_e_* is the adsorbed amount per mass of adsorbent at equilibrium (mg/g). *K_f_* is the Freundlich adsorption constant (L/mg). The magnitude of the 1/n value indicates the strength of the effect of concentration on the adsorption amount.

### 2.6. Specific Adsorption Capacity Experiment

#### 2.6.1. Model Solution Preparation

Model A was the pure gallotannin standard solution used to study the specific adsorption effect of MIPs. Three plant-derived tannins (green tea tannins, grape seed tannins, and grape skin tannins) were used to construct the wine model, and the extraction method was followed as reported by Cheng et al. [25]. The model constructed with green tea was dominated by monomeric tannins and oligomeric tannins, with a low proportion of acylated tannins; grape seeds were dominated by oligomeric gallotannins; grape skin tannins were dominated by polymeric tannins.

Model A: EGCG and ECG were added to the hydroalcoholic solution (ethanol 12%, pH 3.3) at a concentration of 1:1 to make the final concentration of 0.30 g/L.

Model B: Green tea tannins (86.37% purity) were shaken and dissolved in hydroalcoholic solution for a final concentration of 0.30 g/L.

Model C: Grape seed tannins (81.34% purity) had a concentration of 0.30 g/L.

Model D: For Grape skin tannins (88.15% purity), the sample preparation process was the same as the above Model B.

#### 2.6.2. Competitive Adsorption Experiments

MIPs (30 mg) were mixed with a 10 mL model tannin solution, respectively, to determine the specific adsorption capacity of MIPs. After incubation for 4 h at 25 °C, the samples were centrifuged and filtered, and the concentration of free flavan-3-ols monomer in the supernatant was determined by HPLC (Agilent HPLC 1260, Memphis, TN, USA).

#### 2.6.3. Tannin Characterization

The mean degree of polymerization (mDP) and gallotannins (%G) of tannins were found. The polymerization degree was determined by benzyl mercaptan pyrolysis [27]. Briefly, 500 μL of model tannin solution and 500 μL of 5% benzyl mercaptan hydrochloric acid methanol solution (0.2 mol/L) were mixed into a 1.5 mL sample bottle, incubated at 55 °C for 30 min in a water bath, cooled at 4 °C to terminate the reaction, and then analyzed by HPLC (Agilent HPLC 1260) with a VWD detector and Waters XBridge Shield RP18 3.5 µm 4.6 × 250 mm Column. The HPLC was run with a flow rate of 0.6 mL/min at 25 °C. The injection volume was 10.0 μL, and the detection wavelength was 280 nm. The mDP was calculated according to Equation (6), as follows:(6)$$\mathrm{mDP}=\frac{\left(T+E\right)}{T}$$
where *T* represents the percentage of the terminal unit, and *E* represents the percentage of the extension unit.

The %G was calculated using the following formula, Equation (7), with minor adjustments to the algorithm of gallotannins percentage:%G = EGCG% + ECG% + ECG-P% (7)
where EGCG and ECG represent the percentage of free-state monomer, and ECG-P represents the percentage of bound-state ECG.

### 2.7. Color Parameter

CIELab parameters (a*, b*, L*) were measured using a colorimeter (SC-80C, Kangguang, Beijing, China), and pure water was used as the blank sample, according to Obreque-Slier and Ayala [28]. Color intensity (CI) and hue were obtained by measuring absorbance at 420 nm, 520 nm, and 620 nm using an ultraviolet spectrophotometer (UNIC UV2000A, Shanghai, China). All analyses were performed in triplicate.
CI = Abs420 nm + Abs520 nm + Abs620 nm (8)
Hue = Abs420 nm/Abs520 nm(9)

### 2.8. Mitigation Effect of MIPs on Fluorescence Quenching

Fluorescence quenching was performed using the BSA-based method described in Ye et al. [29]. A fluorescence spectrophotometer (Shimadzu-RF-6000, Kyoto, Japan) was used to measure the fluorescence spectra. The excitation wavelength (λ_ex_) was 280 nm; emission excitation (λ_em_) was recorded from 285 to 450 nm, and the slit width was 5 nm. Each model tannin solution was mixed with the BSA protein solution (9 mL, C_protein_ = 1.0 × 10^−6^ mol/L). All samples were incubated at 25 °C for 5 min, and then they were assessed in triplicate.

### 2.9. Electronic Tongue

The corresponding detection parameters for the electronic tongue were based on the method of Han et al. [30], with minor modifications. The ASTREE II e-tongue system (Alpha M.O.S., Toulouse, France) with seven liquid cross-selective sensors (ZZ, AB, GA, BB, CA, DA, and JE) and one reference electrode (Ag/AgCl) was used to test samples. As the recognition of taste relies on the interaction between the molecules and the molecular membrane of the sensor, there is a cross-talk in the perception of taste, which further leads to the fact that the recognition of taste by the electronic tongue cannot be achieved by one sensor alone. In the initialization phase, sensors were calibrated and checked before the sample test. Subsequently, baseline stability was reached after the sensor was rinsed in cleaning water for 120 s. The samples were equilibrated for 30 s before measurement, and the acquisition time was fixed at 120 s. Each sample was measured 10 times, and the data were taken from the 8th to the 10th time. Sensors were rinsed with distilled water between each measurement. The identification and quantification of astringency were based on partial least square (PLS).

### 2.10. Sensory Evaluation

The sensory panel consisted of 7 people (comprising two men and five women, aged between 22 and 60 years old). This panel training method refers to previously reported research [31,32] with minor modifications. All members were trained for astringency, and the specific process is as follows: (i) Utilizing tannic acid to prepare solutions with concentrations of 0.1 g/L, 0.2 g/L, 0.3 g/L, 0.4 g/L, and 0.5 g/L, the astringency intensity was recorded as 1, 2, 3, 4, and 5. (ii) A triangular test was used to determine whether volunteers can perform formal sensory tests.

The assessing protocol started with mouth rinsing with spring water to remove food debris and maintain a ‘neutral’ mouth state. All samples were numbered with random 3-digit numbers, and each sensory round was carried out using a questionnaire with a 5-point scale of astringency intensity. Entire samples were left in the mouth for 12 s, followed by expectorating and re-rinsing the mouth for 12 s before proceeding to the next sample, which was served in 50 mL cups. In total, 3 replicates of each sample pair were assessed over the course of the experiment. Informed consent was obtained for the sensory experiment, and the privacy of the volunteers was strictly protected.

### 2.11. Statistical Analysis

Principal component analysis (PCA) was calculated with a mean score (*n* = 3) of the content of flavanol monomers. The heat map and PCA were conducted with R studio version 4.0.5 (R Studio: Integrated Development for R., Boston, MA, USA), and all other data visualizations were made by Origin 2020 (OriginLab, Northampton, MA, USA). The analysis of significant differences between means was determined by one-way ANOVA using SPSS (Version 22, SPSS Inc., Chicago, IL, USA).

## 3. Result and Discussion

### 3.1. Characterization of MIPs

Scanning electron microscopy: Chitosan had a scaly structure (Figure 1a). There are no pores on the surface of N-MIPs (Figure 1b); thus, the functional groups (-NH_2_ and -OH) of the chitosan shell were connected to the phenolic hydroxyl group by hydrogen bonds or electrostatic force, which is also the reason for non-specific adsorption. The surface of I-MIPs and II-MIPs was smoother (Figure 2C,E) after the template molecules were attached to the surface, while the surface of microspheres after the template molecules were eluted showed folds (Figure 2D,F), which were pores left by the elution of the template molecules. Although the surfaces of I-MIPs and II-MIPs have similar hole-like structures, according to the classical ‘lock-and-key’ theory, different template molecules will cause changes in the structure of functional monomers.

Fourier transform infrared spectroscopy analysis: MIPs (I-MIPs, II-MIPs, N-MIPs) have similar absorption peaks with CTS at 1132 cm^−1^ (C-O), 1568 cm^−1^ (N-H), and 1628 cm^−1^ (C=O), indicating that Fe_3_O_4_ was wrapped by CTS (Appendix A). The absorption peaks at 3521 cm^−1^ of the N-MIPs and CTS were overlapping peaks of the N-H and O-H stretching vibrations of chitosan, but the absorption peaks of the I-MIPs and II-MIPs disappeared, implying that the chitosan was modified on the surface of these two molecularly imprinted polymers. All MIPs had absorption peaks around 1687 cm^−1^ (C=N), which is the characteristic peak for the cross-linking reaction of glutaraldehyde with chitosan. Different from CTS and N-MIPs, infrared spectroscopy showed that I-MIPs and II-MIPs had the C-O characteristic absorption peaks of EGDMA at 1216 cm^−1^, illustrating that the cross-linking agent EGDMA had been grafted on the MICMS surface (Appendix A).

### 3.2. Adsorption

#### 3.2.1. Adsorption Kinetics

Figure 2A shows the curves of the adsorption of MIPs on EGCG with time. The adsorption of polyphenols by the MIPs increased gradually with time, with a rapid increase within 0~100 min, and remained stable after around 240 min. In theory, the initial high-adsorption capacity of MIPs was attributed to an abundance of adsorption sites on the surface, while later spatial site resistance led to slower adsorption rates. The adsorption capacity of I-MIPs reached 76.15 mg/g, which was 12.54 mg/g higher than that of N-MIPs. The adsorption capacity of II-MIPs was about 66.24 mg/g, which was between I-MIPs and N-MIPs, indicating that II-MIPs had a specific recognition site for EGCG, but it was less than I-MIPs.

Only one factor determines the sorption speed, namely the quasi-primary kinetics, while the quasi-secondary kinetic model is based on the assumption that the sorption speed is influenced by the chemical mechanism. The results of the quasi-primary and quasi-secondary kinetic curve fit are shown in Figure 2B,C. Table 1 summarizes the parameters Q_e_ and R^2^ of the kinetic fit equations. The predicted sorption capacity of 75.46 mg/g for I-MIPs was close to the actual measured value of 75.20 mg/g. Interestingly, the quasi-primary kinetic fit for N-MIPs had an R^2^ = 0.997, which was higher than the quasi-secondary kinetic R^2^ = 0.979, indicating that the adsorption process of N-MIPs was more consistent with physical adsorption. The quasi-first-order kinetic fits for I-MIPs and II-MIPs (R^2^ = 0.973, R^2^ = 0.966) were lower than the quasi-secondary kinetic results (R^2^ = 0.982, R^2^ = 0.987), demonstrating the presence of chemisorption for I-MIPs and that II-MIPs were chemisorbed. Therefore, it can be inferred that the recognition of target molecules by I-MIPs and II-MIPs may be chemisorption.

**Figure 2 foods-12-03331-f002:**
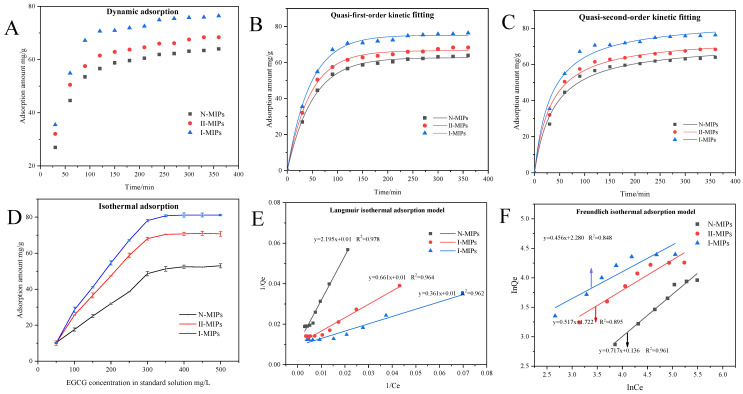
Adsorption kinetics and the isothermal adsorption study: (**A**) adsorption kinetic curve; (**B**) quasi-primary adsorption kinetic fit; (**C**) quasi-secondary adsorption kinetic fit; (**D**) isothermal adsorption; (**E**) Langmuir adsorption model fit; (**F**) Freundlich adsorption model fit.

#### 3.2.2. Isothermal Adsorption

The ability of MIPs to adsorb polyphenols increased with increasing substrate concentration (Figure 2D) and saturated after the substrate concentration reached 300 mg/L. The surface template molecular recognition pores of I-MIPs and II-MIPs increased their ability to adsorb gallic acylated tannins, and the corresponding adsorption capacity was ranked as I-MIPs > II-MIPs > N-MIPs. -MIPs > N-MIPs.

The fitting of the Langmuir isothermal adsorption model to the Freundlich isothermal adsorption model can further determine whether the adsorption of polyphenols by MIPs is unimolecular layer adsorption or multimolecular layer adsorption. Figure 2E,F shows the results of fitting the Langmuir isothermal adsorption model to the Freundlich isothermal adsorption model, respectively, and the corresponding parameter calculations are showed in Table 1. The R^2^ values of the Langmuir models for N-MIPs, I-MIPs, and II-MIPs were 0.973, 0.969, and 0.963, respectively, while the R^2^ value of the Freundlich isothermal adsorption model was 0.963. In general, the R^2^ values of the Langmuir model were higher than those of the Freundlich model, indicating that the adsorption sites on the surface of the MIPs were homogeneous, and when the monomolecular layer on the surface was enriched with the target molecules in the solution, the adsorption sites on the surface became saturated with the target molecules. This indicates that the adsorption sites on the surface of MIPs are homogeneous, and when the monolayer on the surface is enriched with the target molecules in the solution, the adsorption sites on the surface become saturated and belong to the monolayer adsorption.

### 3.3. Specific Adsorption Studies in Model Wines

In this work, four model tannin solutions were prepared. Model A contained EGCG and ECG (%G = 100%); Model B was prepared by the addition of green tea tannins, which were present as monomer phenols (%G = 18.98, mDP = 1.37); Model C was made with grape seed tannins, which were mainly oligomeric gallotannins (%G = 35.45, mDP = 2.94); Model D was made with grape skin tannins with 22.20% gallotannins, and the average degree of tannin was 4.35.

Table 2 showed the percentage of polyphenolic substances adsorbed by MIPs. The α value (selection parameter) was calculated for the increased selective sorption of gallotannins by I-MIPs or II-MIPs compared to N-MIPs, and the corresponding equation was calculated as follows:α = (%G_MIPs_ − %G_N-MIPs_)/%G_N-MIPs_
(10)
where %GMIPs stands for the percentage of I-MIPs or II-MIPs adsorbed to gallotannins; %GN-MIPs is the percentage of N-MIPs adsorbed to gallotannins.

In Model A, there was no significant difference in the percentage of adsorption of EGCG and ECG by N-MIPs (55.14% vs. 44.86%). A total of 68.54% of EGCG was adsorbed by I-MIPs, which was an increase of 13.4% compared to N-MIPs, indicating that the enrichment capacity of I-MIPs for the target substances was enhanced. Although II-MIPs were prepared as composite template molecules (1:1 mol/mol), their adsorption capacity for EGCG was higher than that of ECG, which may be related to the fact that EGCG possesses two gallotropic groups.

The adsorption capacities of I-MIPs and II-MIPs for gallotannins in Model B were 58.73% and 52.78%, respectively, which were 31.21% and 25.26% higher compared to N-MIPs. The reduction of Model B tannin polymerization by N-MIPs (0.16) was higher than that of I-MIPs (0.09) and II-MIPs (0.13), indicating that the adsorption capacity of N-MIPs for polymerized tannins was higher than that of monomers.

The adsorption capacity of N-MIPs for ECG-P in condensed tannins in Model C was 19.18%, which was significantly higher than that for monomers EGCG (2.14%) and ECG (4.28%), and the reduction in tannin polymerization was higher than that of I-MIPs and II-MIPs, further indicating that the adsorption capacity of chitosan molecules on the surface of N-MIPs for high molecular weight tannins was higher than that of the monomers. The adsorption capacities of I-MIPs and II-MIPs for gallotannins in Model C were 71.00% and 56.86%, which were significantly higher than those of Model B. Therefore, the specific recognition sites of I-MIPs and II-MIPs also had adsorption capacities for condensed tannins containing gallotannin groups.

In the polymeric tannin system (Model D), the adsorption of I-MIPs (13.65%) and II-MIPs (15.14%) to ECG-P at the extended end was close to that of N-MIPs (16.64%), while the adsorption capacity to gallotannin monomers was 47.06% and 34.21% higher than that of N-MIPs. The adsorption capacity of I-MIPs and II-MIPs was increased compared to N-MIPs for both gallotannin monomers and polymerization, and their adsorption capacity for monomers was approximately 6.0 times higher than that of polymeric tannins.

Interestingly, in Models B, C, and D, the selective adsorption of gallotannins by II-MIPs was lower than that of I-MIPs, presumably due to competition between the composite template molecules during the formation of the molecularly specific recognition ‘pore’, resulting in the ineffective formation of functional monomers in the cavity. In Model C and Model D, the α values were stable at around 1.80 for the I-MIPs and around 1.30 for the II-MIPs.

### 3.4. Influence of MIPs on the Color Quality of the Model Wine Tannin

This work also studied the influence of MIPs on the color parameter of model wines (Appendix A). The score plot of PC1 versus PC2 showed a distinct clustering of the samples that were related to wine color space, as well as individual deviating samples and groups of deviating samples (Appendix A). The contribution of EGCG and ECG in Model A to the solution color was minor; therefore, the effect of MIPs on the color of Model A was not investigated. In Model B (Appendix A), PC1 explained 53.0% of the variation in the sample set, while PC2 explained 36.05%. All treatment groups were closer to Model B, indicating that the effect of MIPs on the color of Model B was not significant. Further comparison showed that N-MIPs, located towards the positive end of PC1 as well as towards the negative end of PC2 (Appendix A), had higher effects on the color of Model B than I-MIPs and II-MIPs. A similar pattern was seen in Model C (Appendix A) and Model D (Appendix A), where the N-MIP-treated samples had the greatest effect on the CI of the control samples, leading to a lighter color in the solution. The color of the wine was mainly determined by the grape skin tannins, and in Model D, the N-MIPs were further away from the control than in Model B and Model C, as the N-MIPs were more able to adsorb polymeric pigments from the grape skins. Overall, the N-MIPs had the greatest effect on the CI of the model wines, while the I-MIPs and II-MIPs had no significant effect on the color of the wines.

### 3.5. Analyze the Impact of MIPs on Astringency

#### 3.5.1. Fluorescence Spectrum Analysis

The BSA fluorescent signal was quenched due to the masking of the fluorescent group, such as tyrosine, tryptophan, and phenylalanine, and higher quenching effects represent stronger interactions and astringency intensity. Figure 3 reflects the quenching ability of the model tannin solution after MIP treatment on BSA fluorescence. The ability of I-MIPs to attenuate astringency was highest in Model A, which was not unexpected because I-MIPs adsorbed more EGCG and ECG than II-MIPs and N-MIPs in Model A (Table 2). A similar ranking of fluorescence signal reduction capacity was found in Model B: I-MIPs > II-MIPs > N-MIPs. It is interesting to note that in Model C, the fluorescence signal was higher in the N-MIP treatment group than in the II-MIPs, which may be related to the increased proportion of condensed tannins in the model. It is noteworthy that the fluorescence signal of the I-MIP group was still higher than that of the N-MIPs in Model C, even though the N-MIPs reduced the tannin polymerization of Model C by 0.72, whereas the I-MIPs only reduced it by 0.33 (Table 2), probably related to the stronger astringency intensity of gallotannins. N-MIPs showed the highest fluorescence signal peak in Model D, further confirming that the tendency of the N-MIPs to adsorb high levels of tannins led to a significant reduction in astringency. In Model D, N-MIPs showed the highest fluorescence signal peak, further confirming that the tendency of N-MIPs to adsorb high levels of tannins resulted in a significant reduction in astringency. In conclusion, I-MIPs had a greater potential for astringency reduction in the mono/oligomeric tannin models (Model A, Model B, and Model C), while N-MIPs had a greater potential for astringency reduction in the high polymeric tannin models.

#### 3.5.2. Electronic Tongue Analysis

The recognition of taste by the electronic tongue is based on the identification of potential signals from multiple sensors to substances with astringency, and the use of partial least squares regression to draw standard curves remains the more accepted method of study. Tannic acid was used as the astringency standard and the corresponding electronic tongue sensor signal values are shown in Appendix A. As shown in Appendix A, the relative standard deviation values of all samples were <1.6 %. This suggests that the assay variation of the sensors was minimal and that reproducible results could be generated. The results for different concentration gradients of tannic acid and samples are shown in Appendix A, with the tannic acid samples concentrated on the negative half-axis of the t2 axis. A fitted curve was created with the t1 axis as x and the concentration as y (Appendix A), and the corresponding results are as follows:(11)$$y=0.0011e^{-x/0.46}+0.15$$

The astringency of Model A, Model B, Model C, and Model D increased in order, reaching astringency levels of approximately 130 mg/mL, 153 mg/mL, 241 mg/mL, and 307 mg/mL tannic acid equivalent (Figure 4A), respectively. Reduction trends in astringency by MIPs were similar to the fluorescence results. In the four model solutions, I-MIPs reduced the astringency intensity of the Model C solution most significantly, reaching about 30 mg/L tannic acid equivalent of astringency intensity, indicating that I-MIPs are more suitable for reducing the astringency intensity of oligomeric gallotannin-rich wines. Although II-MIPs and N-MIPs exhibited the highest astringency reduction in Model D, the mechanisms were different. II-MIPs showed greater adsorption of condensed gallotannins compared to I-MIPs, due to the higher astringency intensity of condensed tannins, which also resulted in II-MIPs exhibiting higher astringency reduction in Model D than I-MIPs. N-MIPs reduced the astringency intensity of Model D by approximately 32 mg/L tannic acid equivalent, which was driven by the removed condensed tannins.

#### 3.5.3. Sensory Analysis

Similar to the predicted results of the electronic tongue, the astringency intensities of Models A and B were similar (Figure 4B), and the astringency intensity of the I-MIPs treatment group was the weakest, while the N-MIPs were not as effective as the I-MIPs and II-MIPs in diminishing astringency. However, the astringency of Model C predicted by the electronic tongue was lower than that of Model D because the identification of tannin gallate groups by the electronic tongue was not considered to contribute to the astringency intensity. Overall, the sensory results were similar to the electronic tongue and fluorescence results, with Model A and Model B ranking I-MIPs > II-MIPs > N-MIPs. N-MIPs, on the other hand, had an increasingly significant astringent weakening function in the oligomeric and polymeric tannin models.

## 4. Discussion

The absorption capacity of different clarification agents to polyphenols mainly includes chemical driving force and physical driving force. For protein clarification agents, the proline-rich protein has one of the highest affinities for polyphenols in protein clarification agents, and the chemical bond between proline-rich amino acid residues and polyphenols is mainly hydrogen bonds [8]. PVPP, a non-protein clarifier and often performed at the must or fermentation stage, has non-specific adsorption to polyphenols. The presence of micropocket structure on the surface of PVPP microspheres promotes it to interact and capture large polyphenols. Moreover, the small cavities created by the porous structure of PVPP are conducive to capturing and retaining the smaller polyphenols. The higher sensibility of MIPs mainly depends on the specificity of the recognition sites that exist on the surface of the template molecules [33]. The chemical force-driven binding has a better recognition ability than the physical adsorption capacity, and the specific recognition receptors on the surface of MIPs make them have higher specificity than traditional clarifying agents.

Proteins of animal or plant origin are the most common adsorbents in wine. Protein adsorbents have a higher affinity for larger molecular tannins and have less influence on polyphenols with lower molecular weights. Maury et al. [34] found that higher molecular weights of gelatin can absorb more phenolic substances, and they have a higher affinity for polymeric phenols. The adsorption of ovalbumin to polyphenols mainly depends on the volume of the protein, which has no specificity at the molecular level. Jauregui et al. [7] thought that 0.10 mg/mL of whey protein would reduce the astringency of wine from 160 mg/L to 120 mg/L (tannic acid equivalent). Electronic tongue experiments showed that MIPs (3.0 mg/mL) effectively reduced the astringency of the model tannin solution by 30 mg/L (tannic acid equivalent). Compared with traditional adsorbents, the total adsorption capacity of I-MIPs for polyphenols is 81.17 mg/g (EGCG equivalent), which is about one-tenth of the same quality traditional absorbents. In the complex model wine solution, the specific adsorption capacity of I-MIPs to gallotannins can reach 73.0%.

The influence of all clarifying agents on polyphenols should be reflected in the organoleptic qualities, especially the weakening of astringency, and to some extent should affect the sub-quality astringency. Our work showed that MIPs have a good prospect in adjusting the taste of wine. In the next work, we will deeply explore the regulation of MIPs on astringency subqualities. In addition, the synthesis mechanism of MIPs showed that the key to improving the adsorption capacity is in how to increase the specific surface area of MIPs more efficiently. Therefore, MIPs must become smaller and more uniform particles and have a larger surface area.

## 5. Conclusions

This work used molecular imprinting techniques to specifically adsorb gallotannins from wine and explored the effect of single and composite template molecules on the adsorption of polymer tannins. The “pores” on the surface of MIPs are the structural basis for their specific recognition of gallotannins. Moreover, MIPs have a chemically driven single-molecule layer adsorption on the substance. II-MIPs prepared as composite templates had a lower adsorption capacity for monomeric gallotannins than the I-MIPs prepared as single template molecules. In the four tannin models, the adsorption capacity of I-MIPs and II-MIPs to gallotannins was significantly higher compared to N-MIPs. In the model solutions enriched with monomeric/oligomeric gallotannins, I-MIPs and II-MIPs had a significant ability to reduce the astringency, while having no significant effect on the color of the model wines. N-MIPs, on the other hand, were suitable for the high polymeric tannin-rich model wine system and had the largest effect on the color parameters CI of the model wine. The results showed that the molecular imprinting technique could specifically eliminate a certain type of tannin from the wine to reduce astringency while having a low impact on the color quality of the wine.

The advantage of MIPs compared to conventional-type adsorbents is that the target molecule can be recognized based on the functional group information of the template molecule. Therefore, this adsorption mechanism provides excellent targeting adsorption abilities for MIPs, but at the same time, this also provides the ability for tannins with similar functional group structures to be recognized, for example, the adsorption of polymeric gallotannins by MIPs due to the lack of recognition of the C4-C8/C4-C6 bonding forces present between polymeric tannins in MIPs. The MIPs in this work were Fe_3_O_4_-based chitosan microspheres, which do not involve expensive experimental reagents. Therefore, MIPs are inexpensive to make and are expected to be industrialized adsorption materials for the targeted regulation of tannins in commercial wines.

## Figures and Tables

**Figure 1 foods-12-03331-f001:**
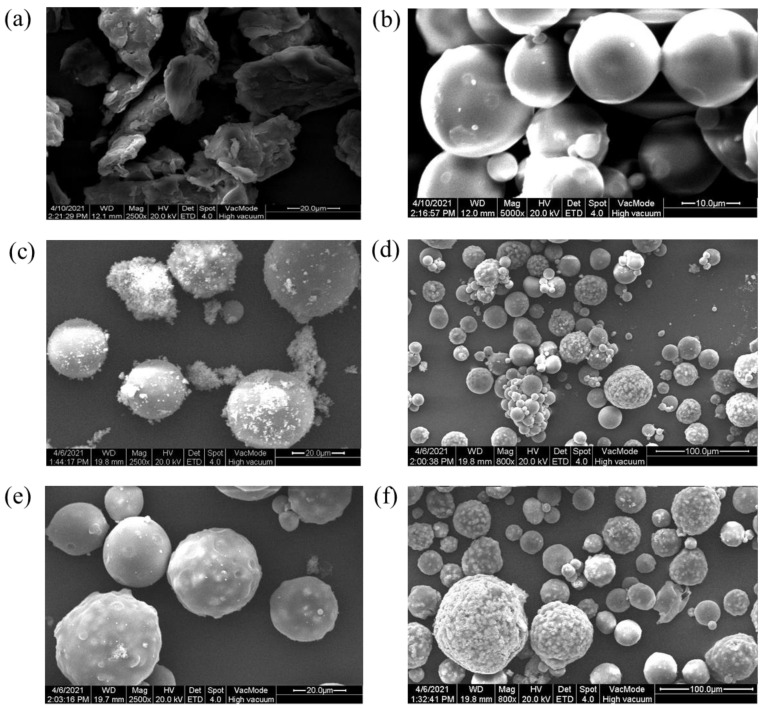
The scanning electron microscopy (SEM) images of MIP material: (**a**) CTS, (**b**) N-MIPs, and (**c**) I-MIPs before the elution of template molecules; (**d**) I-MIPs after the elution of template molecules; (**e**) II-MIPs before the elution of template molecules; (**f**) II-MIPs after the elution of template molecules.

**Figure 3 foods-12-03331-f003:**
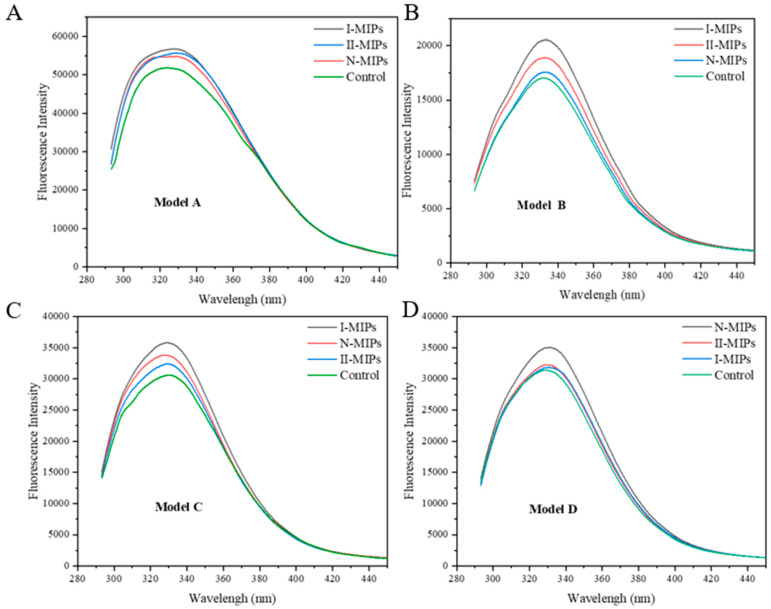
Endogenous fluorescence quenching experiment. Control is the control sample; higher fluorescence intensity means weaker binding of tannins to bovine serum proteins, implying a weaker astringency. Four model systems (Model A, Model B, Model C and Model D), corresponding to (**A**–**D**) graphs, were established to study the targeted adsorption capacity of I-MIPs/II-MIPs/N-MIPs.

**Figure 4 foods-12-03331-f004:**
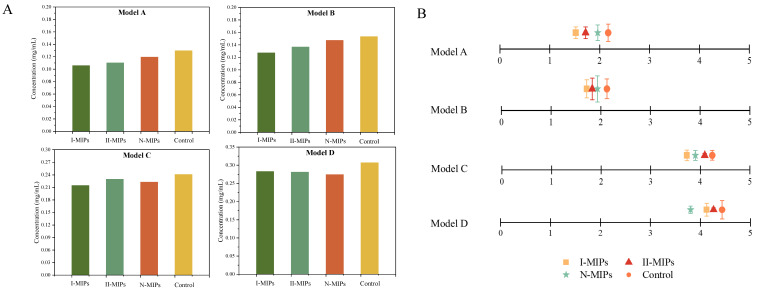
Electronic tongue and sensory verification experiments: (**A**) the prediction of astringency intensity by the electronic tongue was based on the standard curve of tannic acid concentration (Appendix A); (**B**) the sensory results were obtained based on the sensory panel evaluation.

**Table 1 foods-12-03331-t001:** Adsorption kinetic model fitting parameters.

Adsorption Kinetic Model	Parameters *	N-MIPs	I-MIPs	II-MIPs
Quasi-first-order kinetic	*Q_e_*	62.72	75.20	66.64
	R^2^	0.997	0.973	0.966
Quasi-second-order kinetic	*Q_e_*	72.10	75.46	85.18
	R^2^	0.979	0.982	0.987
Langmuir isothermal	*Q_m_*	100.60	99.90	106.84
	*K_d_*	45.83	151.11	295.81
	R^2^	0.973	0.969	0.963
Freundlich isothermal	*Q_m_*	1.15	5.59	0.78
	*K_f_*	1.39	1.94	2.19
	R^2^	0.961	0.895	0.848

* *Q_e_* represents the adsorption value of MIPs at equilibrium. *Q_m_* is the maximum adsorption capacity of the adsorbent (mg g^−1^), *K_d_* is the Langmuir adsorption constant (L mg^−1^), *K_f_* is the Freundlich adsorption constant (L mg^−1^), and R^2^ is the correlation coeffic.

**Table 2 foods-12-03331-t002:** The evaluation of MIPs in four tannin models for the adsorption of gallotannins. Data are expressed as mean ± standard error.

			Adsorption Amount (%) ^1^		
	Parameters ^2^	Control ^3^	N-MIPs	I-MIPs	II-MIPs
Model A ^5^	EGCG	50.00	55.14 ± 1.67	68.54 ± 1.22	60.21 ± 2.31
	ECG	50.00	44.86 ± 0.93	31.46 ± 1.14	39.79 ± 1.12
	ECG-P ^4^	0.00	0.00	0.00	0.00
	mDP	1.00	1.00	1.00	1.00
	%G	100	100	100	100
	α ^9^			0.00	0.00
Model B ^6^	EGCG	15.47 ± 0.89	21.31 ± 1.85	49.21 ± 0.92	36.63 ± 1.14
	ECG	3.51 ± 0.12	6.21 ± 2.34	9.52 ± 0.22	16.15 ± 1.21
	ECG-P	0.00	0.00	0.00	0.00
	mDP	1.37 ± 0.32	1.21 ± 0.14	1.28 ± 0.11	1.24 ± 0.21
	%G	18.98	27.52	58.73	52.78
	α			1.13	0.92
Model C ^7^	EGCG	9.21 ± 1.02	2.14 ± 0.36	45.41 ± 0.47	26.61 ± 0.41
	ECG	7.05 ± 0.33	4.28 ± 0.21	17.43 ± 0.12	20.14 ± 0.31
	ECG-P	19.19 ± 1.27	13.34 ± 1.02	8.16 ± 0.21	10.11 ± 0.11
	mDP	2.94 ± 0.21	2.22 ± 0.11	2.61 ± 0.14	2.49 ± 0.20
	%G	35.45	24.76	71.00	56.86
	α			1.87	1.30
Model D ^8^	EGCG	7.47 ± 0.14	5.17 ± 0.54	42.26 ± 2.14	21.41 ± 0.14
	ECG	4.97 ± 0.24	2.34 ± 0.33	12.31 ± 1.25	20.31 ± 0.14
	ECG-P	9.77 ± 0.64	16.64 ± 0.14	13.65 ± 1.11	15.14 ± 0.11
	mDP	4.35 ± 0.12	3.24 ± 0.22	3.91 ± 0.13	3.67 ± 0.211
	%G	22.20	24.15	68.22	56.86
	α			1.82	1.35

^1^: Adsorption amount is the percentage of MIPs (N-MIPs, I-MIPs, II-MIPs) adsorbed to each type of polyphenol; ^2^: characterization parameters of polyphenols, including EGCG, mDP, %G condensed tannins for extended end-units and end-units; ^3^: percentage of each species of polyphenols in the model solution before treatment with MIPs; ^4^: ECG-P represents the ECG unit of the condensed tannin extension; ^5^: Model A: EGCG+ECG (1:1); ^6^: Model B is green tea tannin, used to simulate wine samples rich in mono/oligo tannin; ^7^: Model C is grape seed tannin; ^8^: Model D is grape skin tannin. ^9^: α was the selection parameter for the adsorption capacity of I-MIPs or II-MIPs.

## Data Availability

The data used to support the findings of this study can be made available by the corresponding author upon request.

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
