# Peer review of "Targeted Removal of Galloylated Flavanols to Adjust Wine Astringency by Using Molecular Imprinting Technology"

_foods, 2023, doi:10.3390/foods12183331_

Round 1

Reviewer 1 Report

The paper by Guorong Du and collegues, entitled “Targeted removal of galloylated-flavanols to adjust wine astringency by using molecular imprinting technology” covers the synthesis of molecularly imprinted polymers (MIPs) with galloylated-flavanols as templates. These MIPs are specifically designed to reduce wine astringency by selectively removing wine polyphenols. The study involved the preparation of three types of MIPs, each based on different templates, and their characterization using various analytical techniques. Furthermore, the MIPs' specificity towards molecules of varying complexity was evaluated by testing them on four wine model solutions with distinct compositions, and the MIPs' performance is evaluated based on several outcomes.

However, some concerns are raised regarding the design of the model solutions used in the experiment. For example, model solutions composition is not clearly described: are the model solutions composed only of 12% ethanol in water with the addition of the three different sets of plant-derived tannins? Moreover, the model solutions did not consider the effect of tartaric acid, known to interact with tannins and proteins in wine, influencing astringency and tannins solubility. Authors should discuss their choice.

Additionally, experimental methods should be described in more detail, as for example, in paragraph 2.5.1 it is not clear how the sampled aliquots were analysed, or paragraph 2.6.3 did not describe the type of HPLC column used for tannin characterisation.

In paragraph 3.2.1, starting from line 270, the absorption capacity is expressed non-uniformly as mg/g and as mg/mL.

The English language is fine; just a few minor spell checks are required.

Author Response

  1. For example, model solutions composition is not clearly described: are the model solutions composed only of 12% ethanol in water with the addition of the three different sets of plant-derived tannins? Moreover, the model solutions did not consider the effect of tartaric acid, known to interact with tannins and proteins in wine, influencing astringency and tannins solubility. Authors should discuss their choice.

Response: Thank you. As your comments, the model solutions composed only of 12% ethanol in water with the addition of the three different sets of plant-derived tannins. Tartaric acid was excluded from all models to diminish the effect of tartaric acid on the astringency of the model wine.

  1. Additionally, experimental methods should be described in more detail, as for example, in paragraph 2.5.1 it is not clear how the sampled aliquots were analyzed, or paragraph 2.6.3 did not describe the type of HPLC column used for tannin characterization.

Response: Thank you for your comments. The EGCG detection method was consistent with the polymerization method described in 2.6.3. The type of HPLC column we used was Waters XBridge Shield RP18 3.5 µm 4.6×250mm Column. (Please see the revised manuscript at line 171).

  1. In paragraph 3.2.1, starting from line 270, the absorption capacity is expressed non-uniformly as mg/g and as mg/mL.

Response: Thank you for your comments. We revised the unit in the manuscript to mg/g. (Please see the revised manuscript at line 265).

Reviewer 2 Report

The purpose of the article and its significance is not stated clearly in the abstract or introduction. The novelty of the study is less well-detailed.

Avoid repetition and try to focus on the main ideas regarding the results. At the same time, the Results and Discussions section could be improved by studying other papers in the field.

Sensory analysis- The sample size is low and may give biased results.

The Conclusion part could be further developed to sharpen the presentation of the specific research deliverables and expected impact in the wine industry. It is important to clearly state the implications for practice.  It did not provide any significant management recommendations.

I suggest including information regarding advantages, limitations, costs, etc., and a brief comment by the Authors on the strengths and weaknesses, this will increase the manuscript's value.

The paper needs some minor revisions for the English language.

Author Response

Reviewer #2:

  1. The purpose of the article and its significance is not stated clearly in the abstract or introduction. The novelty of the study is less well-detailed.

Response: Thank you for your comments, and your opinions were very valuable. We rewrote the research purpose of the article in the abstract and the introduced final section to make the innovative aspect of the article clearer. (Please see the revised manuscript at line 28-30, 69-71).

  1. Avoid repetition and try to focus on the main ideas regarding the results. At the same time, the Results and Discussions section could be improved by studying other papers in the field.

Response: Thank you for your comments. We optimized the views in the results section and added additional discussion about the comparison of the present experimental material with other research results in the field. (Please see the revised manuscript at line 452-454, 465-469).

  1. Sensory analysis- The sample size is low and may give biased results.

Response: Thank you for your comments. The sensory panelists in the experiment were trained in the experiment and the sensory samples were replicated three times to ensure that the results of the sensory experiment were accurate. Hopefully our explanations can allay your concerns.

  1. The Conclusion part could be further developed to sharpen the presentation of the specific research deliverables and expected impact in the wine industry. It is important to clearly state the implications for practice.  It did not provide any significant management recommendations. I suggest including information regarding advantages, limitations, costs, etc., and a brief comment by the Authors on the strengths and weaknesses, this will increase the manuscript's value.

Response: Thank you for your comments. We briefly discuss the advantages, disadvantages, and costs of MIPs materials in the concluding section, which was critical to the article quality. (Please see the revised manuscript at line 497-505).

Reviewer 3 Report

Minor remarks

·         The authors should correct the following errors in the manuscript:

Line

Now it is

It should be

83

…dropwise, quickly stirred (800 rpm) 2.5h.

dropwise, quickly stirred (800 rpm) for 2.5h.

98

Fourier infrared spectroscopy

Fourier transform infrared spectroscopy

112

Adsorption properties refers to

Adsorption properties refer to

129

Mix 30mg MIPs and 10mL EGCG solution (50mg/L ~ 500mg/L) at 30 ℃.

This sentence should be reformulated.

168

…the reaction, and then analyze by HPLC

…the reaction, and then analyzed by HPLC

247

Figure 1. Microscopic image spectroscopy of…

Figure 1. The scanning electron microscopy (SEM) images of…

147

y = 0.0011?−?0.46⁄+0.15

This equation should be numbered.

·         Authors should standardize units throughout the manuscript (for example, ml and mL).

·         Give the full name of the journal Food Chem.

Moderate editing of English language required.

Author Response

Reviewer #3:

  1. The authors should correct the following errors in the manuscript:

Response: Thank you for your comments. We corrected the mistakes in the statements you marked and optimized all the sentences in this article to hopefully meet your requirements. (Please see the revised manuscript at line 85, 99, 102-103, 113, 131-132, 170, 254, 424).

Reviewer 4 Report

Dear authors,

I find your manuscript very interesting. It gives a new insight in wine astrigency adjustment and targeted targeted removal of galloylated-flavanols. The introduction shortly describes the problem and solution. In materials and methods, all methods used in this study have been described. Results have been presented clearly (high resolution figures accompanied with description). Brief discussion gives insight in other studies regarding this subject.

Considering all this, I have no furhter comment and it is my opinion that this manuscript could be published in this journal.

Best regards

Author Response

Response: Thank you for your comments. It would be a great honor to receive your endorsement and I wish you all the best!

Round 2

Reviewer 2 Report

The manuscript can be accepted for publication